# A Case Series Exploration of Multi-Regional Expression Heterogeneity in Triple-Negative Breast Cancer Patients

**DOI:** 10.3390/ijms232113322

**Published:** 2022-11-01

**Authors:** Qi Xu, Jaspreet Kaur, Dennis Wylie, Karuna Mittal, Hongxiao Li, Rishab Kolachina, Mohammed Aleskandarany, Michael S. Toss, Andrew R. Green, Jianchen Yang, Thomas E. Yankeelov, Shristi Bhattarai, Emiel A. M. Janssen, Jun Kong, Emad A. Rakha, Jeanne Kowalski, Ritu Aneja

**Affiliations:** 1Department of Oncology, Livestrong Cancer Institutes, The University of Texas at Austin, Austin, TX 78712, USA; 2Department of Biology, Georgia State University, Atlanta, GA 30303, USA; 3Center for Biomedical Research Support, The University of Texas at Austin, Austin, TX 78705, USA; 4Department of Mathematics and Statistics, Georgia State University, Atlanta, GA 30303, USA; 5University of Nottingham and Nottingham University Hospitals, Nottingham NG7 2UH, UK; 6Oden Institute for Computational Engineering and Sciences, The University of Texas at Austin, Austin, TX 78705, USA; 7Departments of Diagnostic Medicine, Biomedical Engineering, and Oncology, The University of Texas at Austin, Austin, TX 78705, USA; 8Department of Pathology, Stavanger University Hospital, 4011 Stavanger, Norway; 9Department of Clinical and Diagnostic Sciences, School of Health Professions, University of Alabama at Birmingham, Birmingham, AL 35294, USA

**Keywords:** intratumoral heterogeneity, triple negative breast cancer, gene expression

## Abstract

Extensive intratumoral heterogeneity (ITH) is believed to contribute to therapeutic failure and tumor recurrence, as treatment-resistant cell clones can survive and expand. However, little is known about ITH in triple-negative breast cancer (TNBC) because of the limited number of single-cell sequencing studies on TNBC. In this study, we explored ITH in TNBC by evaluating gene expression-derived and imaging-derived multi-region differences within the same tumor. We obtained tissue specimens from 10 TNBC patients and conducted RNA sequencing analysis of 2–4 regions per tumor. We developed a novel analysis framework to dissect and characterize different types of variability: between-patients (inter-tumoral heterogeneity), between-patients across regions (inter-tumoral and region heterogeneity), and within-patient, between-regions (regional intratumoral heterogeneity). We performed a Bayesian changepoint analysis to assess and classify regional variability as low (convergent) versus high (divergent) within each patient feature (TNBC and PAM50 subtypes, immune, stroma, tumor counts and tumor infiltrating lymphocytes). Gene expression signatures were categorized into three types of variability: between-patients (108 genes), between-patients across regions (183 genes), and within-patients, between-regions (778 genes). Based on the between-patient gene signature, we identified two distinct patient clusters that differed in menopausal status. Significant intratumoral divergence was observed for PAM50 classification, tumor cell counts, and tumor-infiltrating T cell abundance. Other features examined showed a representation of both divergent and convergent results. Lymph node stage was significantly associated with divergent tumors. Our results show extensive intertumoral heterogeneity and regional ITH in gene expression and image-derived features in TNBC. Our findings also raise concerns regarding gene expression based TNBC subtyping. Future studies are warranted to elucidate the role of regional heterogeneity in TNBC as a driver of treatment resistance.

## 1. Introduction

Triple-negative breast cancer (TNBC) is a breast cancer (BC) subtype that lacks expression of estrogen receptor (ER), progesterone receptor (PR), and HER2 and accounts for 15–20% of BC cases worldwide. TNBC is a heterogeneous and aggressive disease that displays a grim prognosis with high recurrence and death rates. The aggressive clinical course of TNBC is primarily attributed to the high risk of relapse and metastasis, typically to visceral organs and the brain [1] within the first five years of follow-up. The prevalence of TNBC is particularly high in women of African descent, with TNBC being diagnosed at an earlier age and a more advanced disease stage in African American women than in women of European descent [2]. TNBC has been characterized by extensive inter-patient molecular heterogeneity. Extensive cellular heterogeneity in TNBC clinical specimens can be attributed to the presence of fibroblasts, endothelial and immune cell populations. Immune cells, especially tumor infiltrating lymphocytes (TILs) have recently emerged as clinically relevant biomarkers capable of affecting TNBC prognosis and response to treatment [3,4]. Infiltrating lymphocytes have been strong prognosticators for TNBC patients receiving neoadjuvant or adjuvant chemotherapy as well as for early stage TNBC patients who did not receive any systemic therapy [5]. Stromal cells in tumors are usually represented by the cancer associated fibroblasts (CAFs). Particularly in TNBCs, distinct subsets of myofibroblast- like and inflammatory CAFs have been identified previously that promote cancer cell proliferation and survival [6,7,8]. Although patients with early stage TNBC respond well to chemotherapy, tumor relapse after chemotherapy is frequent [9]. Several clinical trials have demonstrated that compared with patients with non-TNBC, those with TNBC have an increased pathological response (pCR) following neoadjuvant chemotherapy (NACT); however, TNBC patients with residual disease have a significantly worse overall survival than patients with non-TNBC [10]. Although actionable targets, including EGFR, PARP, androgen receptor (AR), FGFR, and angiogenic pathways, are under clinical investigation in TNBC, heterogeneity in clinical outcomes suggests the presence of significant molecular heterogeneity that has not yet been identified, rendering TNBC treatment challenging [9,11].

The two main types of heterogeneity are intertumoral heterogeneity, referring to variation between tumors from different patients [12,13], and intratumoral heterogeneity (ITH), describing regional differences within a single tumor. Advances in single-cell sequencing technologies and analytical methods allow sampling of hundreds of cells from the same tumor specimen to study cellular ITH. Nevertheless, as specimens from different TNBC regions are not typically collected, little is known about spatial ITH in TNBC and its impact on molecular subtyping and pathway enrichment. Prediction of prognosis and treatment decisions are increasingly becoming dependent upon sequencing, and sequencing depends on a representative sampling of a tumor specimen [12]. This representative sampling hinders the understanding of the multi-regional molecular profile of the tumor, which is critical for optimal and accurately informed management of TNBC.

Massive parallel sequencing studies have shown that both spatial and temporal heterogeneities are common in BC [13,14,15,16]. Deep sequencing of tumor-associated somatic mutations has revealed a substantial level of ITH in TNBC, and spatial subclonal diversification is more prominent in TNBC than in other BC subtypes. Recent studies have demonstrated that ITH is a driver of pathogenesis, treatment resistance, metastasis, and poor clinical outcomes. Whole-exome and whole-genome sequencing have revealed that ITH underlies therapeutic resistance and recurrence in TNBC [17,18,19]. Furthermore, although the mutation rate of ER+ tumors was found to be similar to that of normal cells, TNBCs exhibited a 13-fold higher mutation rate [20,21]. Collectively, these findings suggest that TNBC is characterized by persistent intratumoral diversification. Therefore, biopsies of a single tumor region are unlikely to accurately represent the genetic, epigenetic, and phenotypic characteristics of the entire tumor [12,22].

ITH may significantly influence the outcomes of personalized treatments, which commonly rely on a biopsy from a single region of the tumor to represent a patient tumor’s gene expression profiles and cellular composition. Hence, a better understanding of the mechanisms and consequences of ITH, as well as the development of novel approaches to characterize ITH, are crucial for improving the efficacy of personalized anticancer therapies, particularly in TNBC, which has been shown to exhibit profound ITH.

In this study, we analyzed the molecular, phenotypic, and cellular profiles of 34 multi-regional tumor samples from 10 TNBC patients by RNA sequencing (RNA-seq) and quantitative image analysis (Figure 1, top). Our goal was to elucidate the variability in gene expression and cellular density between patients, across regions of different patients, and between regions within the same patient (Figure 1, bottom). Figure 1 displays the distinction between these levels of variation using a simple hypothetical example of a 4-gene signature. The first two categories of variation (i.e., between patients and between-patients across regions) are relative; that is, the same gene signature is examined relative to each patient. In contrast, the last category (i.e., between regions within the same patient) is absolute; thus, the gene signature is specific for each patient. Our novel, distance-based analysis framework for characterizing the different levels of variation and their effects on molecular subtyping and tumor composition revealed that multi-regional sampling in sequencing studies could lead to heterogeneity in marker-based treatment outcomes, prognosis, and etiology [23]. Collectively, these results underscore the importance of multi-regional sampling in TNBC sequencing studies.

## 2. Results

### 2.1. Tumor and Patient Characteristics

The clinicopathological characteristics of the 10 TNBC patients, including age at diagnosis, menopausal status, tumor size, nuclear grade, treatment, recurrence, distant metastasis, date of initial diagnosis, date of surgery, and patient survival status at last contact, are shown in Table 1 (Appendix A). Patients were followed up for a minimum of ~11 years. Adjuvant chemotherapy treatment with cyclophosphamide, methotrexate, and 5-fluorouracil (CMF) was administered to 7 of 10 (70%) patients, none of whom received neoadjuvant chemotherapy. Most patients had large (≥2 cm), grade 3, invasive ductal tumors. Most patients were alive with no distant metastasis at the last follow up, suggesting a good response to CMF treatment.

### 2.2. Between-Patients (Intertumoral) Gene Expression Heterogeneity

We performed gene expression variability analysis using gene expression transformed counts (Appendix A) and identified two gene sets: (1) between-patients—as compared to within-patient gene expression variability across regions (108 genes) and (2) between-patients across regions defined by greater within-patient gene expression variability across regions as compared to between-patient variability (183 genes) (Figure 2A and Appendix A).

Protein-coding genes comprised the majority of each signature gene set (56% for between-patient, 73% between-patient across regions); the remaining genes were pseudogenes. Gene set enrichment analyses revealed a significant (*p* = 0.05) enrichment of the following gene signatures in the between-patient gene expression signature: cholesterol homeostasis, estrogen response early, fatty acid metabolism, and myogenesis. No significant enrichment of hallmark cancer gene sets was identified in the between-patients across regions gene signature. However, this between-patient signature did include the genes *PAXIP1* and *RAD51D* as part of the homologous recombination pathway, although not specifically enriched.

Cluster analysis of the between-patient gene expression signature showed clustering of regions from the same patient and separation into two patient subclusters: C1 with *n* = 4 patients and C2 with *n* = 6 samples; Figure 2B. We examined the relationship between tumor and molecular characteristics in these two subclusters and identified age at diagnosis and menopausal status as significantly (*p* < 0.20) associated with the feature associated with age at diagnosis and menopausal status. Specifically, cluster C1 was enriched in post-menopausal patients diagnosed at an older age; in contrast, cluster (C2) consisted of premenopausal TNBC patients diagnosed at a younger age (Figure 2D). Cluster analysis of the between-patient across regions gene expression signature showed that except for one patient (P6), multiregional tumor samples from the same patient were grouped into different clusters, confirming that this signature signifies intratumoral heterogeneity in gene expression (Figure 2C).

Pathway enrichment analysis showed that significantly enriched pathways among the “between-patient, across regions” signature included the PD-1 and Notch signaling pathways, while ERBB4 signaling, metabolic reprogramming, and multidrug resistance pathways were enriched in the between-patient signature (Figure 3; Appendix A). We further divided the between-patient set (*n* = 108 genes) into gene sets associated with each of the two main patient subclusters and performed pathway enrichment analysis (Appendix A). The between-patient signature (*n* = 45 genes) associated with older age at diagnosis and post-menopausal, stage 2 TNBC was significantly enriched in metabolic reprogramming and multidrug resistance pathways; glycolysis was enriched in both the within-patient signature and the C1 cluster signature. The between-patient signature (*n* = 63 genes) associated with younger age at diagnosis and premenopausal, stage 2 and 3 TNBC was significantly enriched in APC-related pathways, including activator regulation.

### 2.3. Gene Expression Heterogeneity within Patient, between Regions

We analyzed expression-based features (subtyping, immune cell type, immune score, stroma score, and tumor purity) of different tumor regions within each patient based on constructed distances using our divergent analysis approach. We also used this distance-based approach to identify gene expression signatures specific to each patient characteristic of low- and high-expression variability and explored the collective use of these signatures to derive an enriched network of long-term survivors. We analyzed different tumor regions within each patient for PAM50 subtypes and found that four patients (P1, P2, P6, and P9) exhibited divergent PAM50 classification results (Figure 4A). For example, P1 (T1) shows slightly greater than 50% basal subtype, 28% Her2 and 17% Luminal B (Appendix A). In clinical practice, the PAM50 subtype with the largest estimated probability is assigned. In the case of P6, the probabilities for basal and Her2 were very close, making it difficult to determine the PAM50 subtype. The remaining six patients showed convergence for the basal subtype. No significant associations between patient and tumor characteristics and PAM50 classifications were identified. We further applied PAM50 to the mean gene expression among regions within each tumor to define and evaluate the patient-level representation. With the exception of two patients (P6, P9), the basal subtype was identified. In P9, PAM50 luminal B was identified, and in P6, luminal B and basal subtypes were identified with similar estimated probabilities. By evaluating PAM50 subtyping on mean gene expression, the heterogeneity present in P1 and P2 was lost as it was not present in all regions, in contrast to what was observed for patients P6 and P9.

We also investigated TNBC subtypes in different regions of tumors that revealed a divergence in classification (patients P1, P2, P3, P4, P6, P7, P8, P9; Figure 4B). P1 was assigned TNBC subtype Basal-like 1 (BL1) in T1, Basal-like 1 (BL2) in T2, BL1 in T3, and mesenchymal (M) in T4 regions based on their largest significant correlation (Appendix A). Notably, the four divergent PAM50 classifications were a subset of the seven divergent TNBC subtypes.

A significant (*p* = 0.06) association between patient classification (divergent vs. convergent) and CMF chemotherapy (yes vs. no) was identified, as seven of the eight patients with divergent TNBC subtypes received chemotherapy. These findings suggest that TNBC subtyping based on multiregional sampling rather than a single tumor region may more accurately predict response to treatment. This result implies an underlying link between TNBC molecular heterogeneity and treatment outcomes that would have otherwise been missed based on representative sampling of a single region. In contrast, TNBC subtyping based on the mean gene expression among regions showed three samples as unspecified, with two TNBC patients represented in each of BL1, IM, and LAR subtypes; one tumor was classified as M subtype (Figure 4D).

Immune enrichment analysis revealed two main subclusters that differentiated samples based on the abundance of naïve versus memory B cells (Appendix A). Specifically, all regions from patients P6 and P7 (post-menopausal patients with stage 2 TNBC and older age at diagnosis) showed an abundance in naïve B cells. In contrast, memory B cells were abundant in premenopausal patients with stage 2 or 3 TNBC and younger age at diagnosis. Supervised cluster analysis for each tumor revealed profound heterogeneity in immune cell abundance among different tumor regions (Appendix A). For example, macrophages and memory CD4 and B cells were enriched in the entire tumor of P1 but not in individual regions, suggesting a high variability in immune cell type enrichment among tumor regions. By contrast, in patient P2, memory B cells were enriched in all tumor regions (convergent immune enrichment). Patients P1, P3, P4, and P9 showed divergent immune cell enrichment. We found a significant association between divergent versus convergent immune cell enrichment and vital status, recurrence, distant metastasis, and menopausal status (all *p* < 0.10). Divergent immune cell enrichment was associated with increased patient survival, no recurrence or distant metastasis, and premenopausal status.

Gene expression-based stroma, immune score, and tumor purity predictions showed that six patients (P1, P2, P5, P7, P8, and P9) had varying immune and stroma scores among tumor regions (Appendix A); in contrast, tumor purity remained at approximately 75% in all regions and tumors. This result underscores the importance of multiregional sampling within the context of personalized treatment based on the TNBC microenvironment.

We applied our novel divergent analysis to each feature derived from gene expression and imaging data (Figure 5). TNBC patients were classified into patients with high-divergence tumors and those with low-divergence tumors in terms of expression-derived stromal score, molecular subtyping, and TIL abundance (Appendix A). Table 2 shows a summary of the impact of multiregional gene expression heterogeneity and imaging-derived heterogeneity (Tumor/TIL cell counts) and their relationship with clinicopathological characteristics. In Table 2, the ‘divergent association’ row highlights patient and tumor associations (*p* < 0.20) with divergent versus convergent results. Among gene expression features, lymph node stage was significantly associated with divergent tumors, while grade was associated with image-derived abundance of TILs.

We also analyzed absolute, patient-specific gene expression heterogeneity to define high-variance and low-variance gene sets among different regions of the same tumor. First, we evaluated the level of ITH based on the distributions of genetic distance (Appendix A). After applying a Bayesian changepoint model to these genetic distances, we defined patient-specific gene signatures with high and low variability in their expression levels (Figure 6).

These patient-level signatures might broadly represent ‘clonal’ (low heterogeneity) (Appendix A) and ‘subclonal’ (high heterogeneity) variations (Appendix A). Pathway enrichment analysis showed little overlap between patient-level gene signatures and between gene signatures with high and low variance (Figure 6).

Since low variability genes can reflect one of two scenarios, i.e., either low or high expression among regions, we examined expression levels to differentiate between these two scenarios and found that low-variance gene signatures were mainly expressed (Appendix A). By combining all high-variance gene signatures in a network analysis, we identified an enriched network in living versus deceased patients (Figure 7). Notably, the USP17 gene family (including *USP17L11*, *USP17L20*, and *USP17L17*) was highly represented in the protein-coding network.

## 3. Discussion

BCs exhibit substantial phenotypic and genetic intratumoral heterogeneity. In this study, we examined the extent of discordance between multiple tumor samples that originated from the same patient by analyzing the intrapatient and interpatient variance of gene expression, tumor cell density, and TIL density. Multiregional RNA-seq and image analyses of consecutive tumors from the same patient provided evidence of significant intratumoral heterogeneity. Notably, gene expression profiling of multiple samples from the same tumors revealed that the gene expression variance was higher within tumors than between tumors of different patients.

Moreover, we performed TNBC subtyping analysis and found that 70% of the patients exhibited heterogeneous molecular subtypes. The remaining 30% of patients who presented with more homogenous TNBC subtypes had luminal androgen receptor or mesenchymal subtype of TNBC. These findings are in line with previous studies showing that these TNBC subtypes are associated with pCR and favorable overall survival, suggesting low tumor heterogeneity [24,25,26].

Intratumor heterogeneity in TNBCs may also stem from the distinct gene expression profiles of different cell types within TNBC microenvironment. As in our study, we found that PLK1 and Notch signaling pathways were enriched among genes with high within-patient expression variability. Notch signaling plays a critical role in TNBC. Overexpression of JAG-1 and Notch-1 has been associated with poor overall survival in patients with TNBC, and the Notch pathway has been shown to promote TNBC cell proliferation. Previous studies have demonstrated that Notch signaling is associated with the regulation of tumor-initiating cells as well as with the regulation of TNBC etiology. Furthermore, Notch signaling has also been implicated in playing a major role in breast cancer stem cells maintenance and expansion [27,28]. Notch signaling has also been implicated in the development of small-cell lung cancer and has been shown to increase intratumoral heterogeneity [29,30]. Thus, Notch pathway activation in a subset of TNBC cells may contribute to intratumoral heterogeneity, and certain patients with TNBC may benefit from Notch pathway inhibitors in combination with chemotherapy. Furthermore, results from the protein-coding network exhibited a high representation of USP17 gene family. USP17 subfamily genes encoding deubiquitinating enzymes were identified as immediate early genes that can be rapidly induced in response to cytokine stimulation in mice and humans [31]. Specifically, USP17 has been shown to regulate inflammation, cell motility, Th17 cell development, and oncogenesis [32]. Moreover, high levels of USP17 have been shown to promote G1-S transition and cell proliferation in multiple cancer cell types. However, the role of USP17 in BC remains largely unexplored. One study demonstrated that USP17 acted as a tumor suppressor by deubiquitinating asparaginyl endopeptidase, thereby promoting breast cancer development and progression [33]. Further studies are warranted to explore the role of USP17 family genes in BC. 

Immune evasion by tumor cells has been implicated as a hallmark of cancer, wherein tumor cells evade attack and elimination by the immune system [34]. The crosstalk between tumors and immune system is complex, that is, tumor cells not only have the ability to escape immunologic defenses but are also shaped by their immune surroundings, a process called immunoediting [35]. Specifically, TNBCs, TILs have emerged as clinically relevant biomarkers capable of affecting TNBC prognosis and response to treatment [36]. In this regard, it is imperative to conclude that TILs and gene signatures associated with immune cells have important implications for clinical response and hold significant prognostic value. Additionally, the interactions between tumor cells and tumor-infiltrating inflammatory cells can increase phenotypic heterogeneity and may influence treatment response [37,38,39,40]. Our image analysis revealed that the variance in tumor cell and TIL densities within patients was higher among patients with high gene expression variance (Appendix A). In addition, high variance of TIL area was associated with large tumor size, young age, premenopausal status, and high tumor grade. These findings are in line with our gene expression data and collectively suggest that intratumoral cellular heterogeneity is greater than intertumoral heterogeneity.

Transcriptomic profiling studies have shown that BCs can be classified into five intrinsic subtypes: luminal A, luminal B, HER2-enriched, basal-like, and normal breast-like (5, 6). Although all intrinsic subtypes can be found in IHC-defined TNBC tumors, most TNBCs (50–75%) exhibit a basal-like phenotype [9,24,41,42]. Approximately 80% of basal-like tumors are ER-negative/HER2-negative. Although PAM50 intrinsic subtypes have been associated with intertumoral heterogeneity in TNBC, the relationship between PAM50 subtypes and intratumoral heterogeneity remains unclear.

Our study has some limitations. The cohort size was very small, potentially limiting the generalizability of our findings. The uneven distribution of the molecular subtypes of TNBC is another major limitation, as the lack of a larger sample size with known subtypes of TNBC precluded the study of the confounding effect of subtypes. In this study, the resolution of intratumoral heterogeneity was characterized on a macroscopic scale rather than a microscopic scale. Single-cell gene expression profiling studies should be conducted to characterize intratumoral heterogeneity at a higher resolution. Furthermore, the distance between the multiregion samples was unknown. Samples from the border of the tumor, the surrounding stroma, and the sub-border may differ from those from the center of the tumor. Often, the border of the tumor has a higher level of cellularity and more extensive angiogenesis than the core of the tumor. The study provides a new lens on intratumoral heterogeneity from regional sampling. Since it involves looking at intratumoral heterogeneity from vantage point of the regions and not several cells from a single region tissue as is the case in most of the studies. Furthermore, any comparison to single cell results when sampling many cells of the same region tissue or bulk tumor results would not be informative as they are not comparable, and as such, differences in results are likely due to differences in tissue sampling. Moreover, because of the paucity of tumor tissues, we were not able to stain the samples for common biomarkers. Studies have reported extensive heterogeneity in the protein expression of commonly used biomarkers. Future studies involving large cohorts with well-annotated clinicopathological characteristics are required to understand the role of intratumoral genomic and phenotypic differences in patient prognosis and treatment outcomes.

## 4. Materials and Methods

### 4.1. TNBC Samples

Formalin-fixed paraffin-embedded (FFPE) tissue sections and hematoxylin-eosin (H&E)-stained slides of patients diagnosed with TNBC between 1987 and 1998 were obtained from Nottingham City Hospital, UK. The triple-negative status of these cases was confirmed by immunohistochemistry (IHC) as part of the Nottingham-Tenovus Primary Breast Carcinoma Series [43,44]. All cases were histologically reviewed, and diagnoses were confirmed by three independent pathologists. All study aspects were approved by all Institutional Review Boards (Gorgia State University; University of Nottingham and Nottingham University Hospitals) were conducted in compliance with material transfer and data use guidelines of all involved institutions. Written informed consent was obtained from all subjects.

Representative FFPE blocks (*n* = 34 samples) from ten patients with TNBC were retrieved. H&E-stained sections of 21 samples from seven patients were observed microscopically to determine tumor burden (at least 50% tumor burden relative to the tissue cellularity of the entire specimen) and to guide tumor macro dissection. Multiregional samples from these patients were prepared during routine specimen gross examination as per the Royal College of Pathologists guidelines for reporting breast disease in surgical excision specimens [45]. Briefly, specimens were incised with a cruciate incision to embed spatially representative areas of the tumor and to accurately determine the tumor size. Four 10 μm-thick unstained sections were prepared from each block, and the invasive tumor tissue was macro dissected. Macro dissected tissues from each sample were deparaffinized, rehydrated, and centrifuged to remove excess ethanol.

### 4.2. RNA-seq Data Processing

RNA was extracted using the Omega Mag-Bind XP FFPE RNA isolation kit (Omega, Biel/Bienne, Switzerland, M2595-01) and KingFisher Flex magnetic particle separator (ThermoFisher, Waltham, MA, USA). RNA concentration was measured using a Nanodrop 2000c spectrophotometer (Thermo Scientific Inc., Waltham, MA, USA). RNA integrity was assessed using Agilent 2200 TapeStation (Agilent Technologies, Santa Clara, CA, USA), and the percentage of fragments larger than 200 nucleotides (DV200) was calculated. First-strand cDNA synthesis was performed using ~100 ng RNA at 25 °C for 10 min, 42 °C for 15 min, and 70 °C for 15 min, followed by RxnPure magnetic bead clean-up. The final libraries were validated using Agilent High Sensitivity D1000 ScreenTape on an Agilent 2200 Tapestation instrument. The size distribution of the library ranged from ~200 bp to 1 kb. Libraries were normalized, pooled, and clustered. Pair-read sequencing was performed for 75 cycles on a HiSeq2500 instrument (Illumina, Inc., San Diego, CA, USA) according to the manufacturer’s instructions. An index was built on GRCh38.P10 using Salmon [46] index command, and the index alignment-free transcript abundance was determined. Transcript-level abundance was imported into tximport [47] to calculate gene abundance. Batch effects and unwanted variation were eliminated using SVA [48]. DESeq2 [49] was used to obtain relative log expression (RLE)-normalized gene expression levels.

### 4.3. Variability Analysis of Gene Expression

Between-patients versus between-patients across regions. We developed a single statistical model to identify a single gene with varying expression levels between patients and a single gene signature with varying expression levels across regions of different patients. By fitting a linear model to a defined vector of gene expression data within each gene (treating each patient tumor as an independent factor), we calculated two standard deviations (SD): (1) between tumor samples across regions and (2) between patients. The scatter of the two standard deviations was examined, and genes with high intertumoral heterogeneity in their expression were identified based on their high intertumoral to intratumoral SD. The number of genes selected as characteristic of each set was significantly (*p* < 0.0001) greater than the expected number of genes identified by chance due to random sampling. Hierarchical clustering of gene expression data was performed using the R package ComplexHeatmap [50].

Within-patient, between regions (regional intratumoral heterogeneity). A gene-level metric of genetic distance was derived [23] by calculating the Euclidean distance in gene expression among all region pairs within a patient. The genome-wide distribution of genetic distances for all genes was used in a Bayesian changepoint model [51] to define gene sets with high and low variability for each patient.

### 4.4. Enrichment Analysis

Pathway enrichment analysis was performed using enrichR [52] and ClusterProfiler [53]. Gene set enrichment analysis was performed for each gene signature against the hallmark cancer gene set based on a hypergeometric distribution [54]. *p*-values ≤ 0.05 were considered statistically significant. To identify gene networks enriched in long-term survivors (versus deceased patients), we used hypermodules [55] to cluster the ten patients into two sets: a network training set (P1/P3/P4, long-term survivors) and a network validation set (consisting of the remaining long-term survivors and deceased TNBC patients). An input ‘training’ network was built based on the high-variance gene sets identified among the three long-term survivor patients with one-to-one interaction. The validation gene list consisted of high-variance genes among the remaining seven patients (*n* = 3 long-term survivors, *n* = 4 deceased) and was used to identify submodules enriched in long-term survivors. We implemented this approach separately for all high-variance genes and protein-coding genes within the high-variance gene sets.

### 4.5. Association Analysis

Patient-level associations with clinicopathological variables were examined using Fisher’s exact test. A *p*-value ≤ 0.20 was considered statistically significant because of the small sample size.

### 4.6. Subtyping

Gene expression data were expressed as counts per million (CPM) and were log2-transformed. PAM50 [56] subtyping was conducted using the genefu package [57] and was applied to a dataset of breast tumors and normal samples. For tumor samples, we combined our RNA-seq data of 34 multiregional samples from ten patients with 931 TCGA-BRCA primary invasive breast tumor samples with at least 60% tumor purity. For the normal samples, we used RNA-seq data from normal breast tissues from GTEx (*n* = 290) and normal breast tissue samples from the TCGA-BRCA data portal (*n* = 113). The results of PAM50 were expressed as the probability for each sample to classify into each subtype.

TNBC molecular subtyping was performed using TNBC type [58,59] gene expression matrix of 34 samples. In addition, an average TNBC type subtype was defined based on the mean gene expression among all regions within each patient. Gene expression CPM data were log2-transformed before subtyping, and all ESR1 values were set to 0, as all cases were confirmed to be ER-negative based on IHC. The results of TNBC subtyping were expressed as a correlation matrix and confidence intervals between samples and subtypes; a correlogram based on the correlation matrix was created using corrplot [60].

### 4.7. Prediction of Immune, Stroma, and Tumor Purity Scores

We performed CIBERSORT analysis using RNA-seq CPM data to predict absolute immune abundance scores based on the LM22 signature gene sets. We also performed ESTIMATE analysis using RNA-seq to predict stromal, immune, and tumor purity scores.

### 4.8. Slide Annotation

All diagnostic slides of the surgical samples were reviewed by a pathologist. Fresh 5 μm-thick full-face sections were prepared from FFPE blocks, stained with H&E, and reviewed by a pathologist. Two sets of slides were scanned with a digital slide scanner using a 40× magnification objective lens (0.24 μm/pixel; Pannoramic 250 Flash III, 3DHISTECH, Budapest, Hungary) in mrxs format and with the Olympus Nanozoomer whole-slide scanner using a 20× magnification objective lens in ndpi format. The resulting whole-slide images (WSIs) were reviewed using CaseViewer (3DHISTECH Ltd., version 2.3) and ImageScope (version 12.3.2.8013, Leica Microsystems). WSIs with out-of-focus areas were rescanned, and those with folded tissues were excluded from further analysis. Cell centers and different tissue components identified by the pathologists were manually annotated in randomly selected regions of variable size using ImageScope. For training purposes, image patches of 128 × 128 pixels for mrxs and 96 × 96 pixels for ndpi files were generated from the pathologist-identified “ground truth” images. Different image patch sizes were selected from mrxs and ndpi files to accommodate different magnification levels. To count tumor cells and tumor-infiltrating lymphocytes (TILs), we trained one model for each cell type with each type of whole-slide microscopy image. In total, 52,945 and 53,361 image patches were annotated for tumor cells in mrxs and ndpi WSIs, respectively.

### 4.9. Divergent Analysis

To compare the degree of regional heterogeneity in gene expression and image-derived quantitative features between patients, we calculated the Euclidean distance among all region pairs with respect to each quantitative feature (e.g., subtype probability, stroma score, and tumor counts) within each patient. We applied a Bayesian changepoint model [51] to the patient-level distributions of distances to define patients as either divergent (high-valued distances) or convergent (low-valued distances) for each feature. For image features, we used manually annotated averaged tumor counts and TILs (Appendix A).

## 5. Conclusions

In conclusion, our results suggest that TNBCs exhibit higher intratumoral than intertumoral variability in gene expression and cellular densities, suggesting the presence of profound intratumoral heterogeneity in TNBC. These findings suggest that single biopsy specimens may reveal only a portion of genetic aberrations that are present in the entire tumor. These genetic aberrations contribute to tumorigenicity, activation of signaling pathways, induction of senescence, angiogenesis, cancer cell migration, and response to treatment. Therefore, treatments selected based on the molecular profile of diagnostic biopsies may fail to eliminate the bulk of the tumor, ultimately leading to tumor recurrence. Targeting a highly heterogeneous tumor comprising multiple cell clones is clinically challenging. Thus, understanding the molecular mechanisms driving intratumoral heterogeneity in TNBC may help identify novel therapeutic targets.

## Figures and Tables

**Figure 1 ijms-23-13322-f001:**
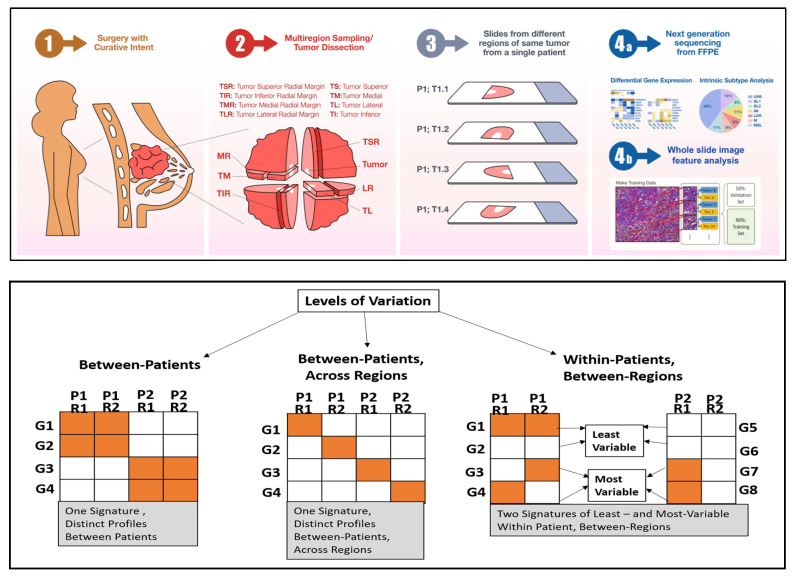
Overview of study design and analyses. (**Top**): Schematic representation of specimen selection, multiregional sampling, data generation, and processing steps. (**Bottom**): Schematic of three levels of variation: between-patients, between-patients across regions, and within-patients between regions. For the first two levels, a single gene signature is defined (based on four genes in this schematic) with distinct expression profiles (orange, expressed; white, not expressed) between patients or between patients and across regions. In the third variation level, 2-gene signatures are defined for each patient; one with high and one with low variation between regions. In this example, the least variable gene signature could be characterized by either expressed genes or genes with low to no expression (white).

**Figure 2 ijms-23-13322-f002:**
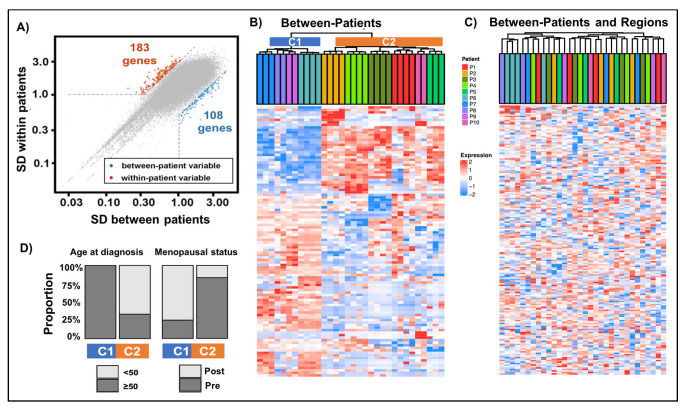
Gene expression heterogeneity between TNBC patients and regions characterizes patients based on age at diagnosis and post-menopausal status. (**A**) Selection of genes with high expression variability between patients and across regions. Scatterplot of standard deviations (SD) of gene expression between patients and between regions. In red are 183 genes with within-patient gene expression variability greater than one and greater than or equal to nine times the estimated between-patient variability. In blue are 108 genes with between-patient expression variability greater than one and greater than or equal to four times the estimated within-patient variability. (**B**,**C**) Unsupervised clustering of genes characteristic of between-TNBC patients versus between-TNBC patients and regions. RNA expression data (log2[CPM + 1]) were scaled in rows and columns and were clustered using Ward’s algorithm based on Euclidean distance. The heatmap of 108 genes characterizing between-patient differences (**B**) shows two main patient clusters: C1 (blue, *n* = 4 patients) and C2 (orange, *n* = 6 patients). The heatmap of 183 genes characterizing between patient and region differences (**C**) shows clusters of regions from the same tumor (with the exception of one patient). (**D**) Distribution of significant (*p* < 0.05) patient and tumor characteristics associated with between-TNBC patient clusters represented by colored bars.

**Figure 3 ijms-23-13322-f003:**
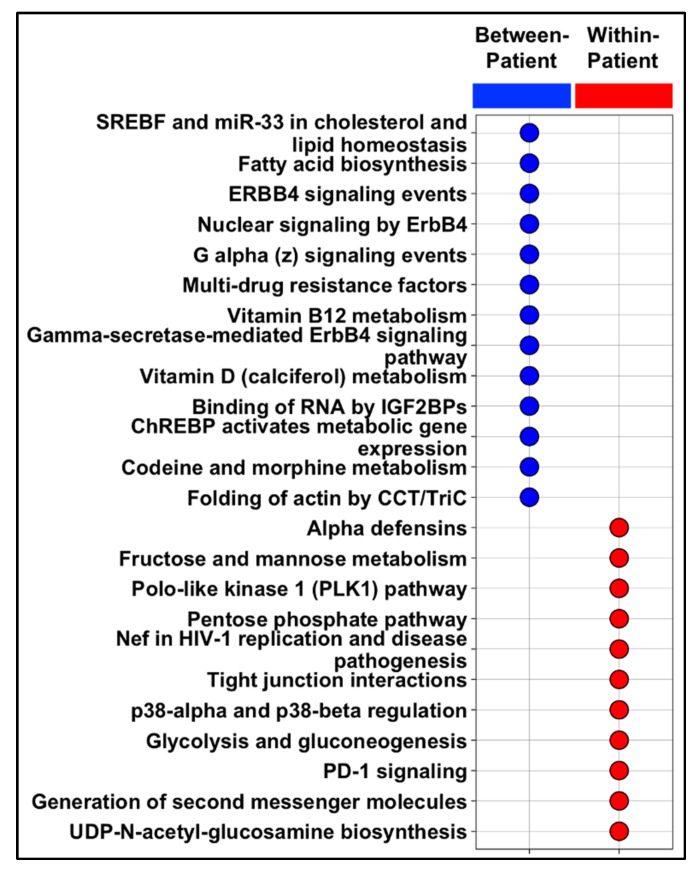
Enrichment analysis shows distinct pathways enriched in gene signatures with high regional ITH and intratumoral heterogeneity. Significant (*p* < 0.05) enrichment of pathways from between-patient gene expression signature (blue, 108 genes) and within-patient signature (red, 183 genes).

**Figure 4 ijms-23-13322-f004:**
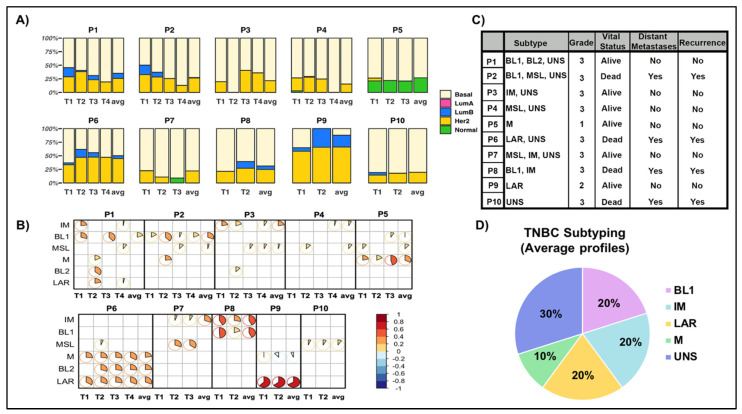
Molecular classification differs among different regions of the same tumor. (**A**) PAM50 subtyping. For each patient (denoted by P) and tumor regions (denoted by T) and average gene expression over regions (denoted by avg), PAM50 subtyping results were expressed as probability to classify into each of the five subtypes: basal-like (basal), luminal A (LumA), luminal B (LumB), HER2-enriched (HER2) and normal-like (normal). (**B**) TNBC subtyping. Correlogram of TNBC subtyping results for each tumor region and average (over regions); significant (*p* ≤ 0.05) results are shown. Each pie chart represents the correlation between the tumor sample with each TNBC subtype, with the direction of correlation as positive (red) and negative (blue). (**C**) Summary of regional TNBC subtype classifications combined with patient and tumor characteristics. BL1: Basal-like 1; IM: Immunomodulatory; LAR: Luminal androgen receptor; M: Mesenchymal; UNS: Unspecified group. (**D**) TNBC subtyping assignment based on mean (over regions) gene expression.

**Figure 5 ijms-23-13322-f005:**
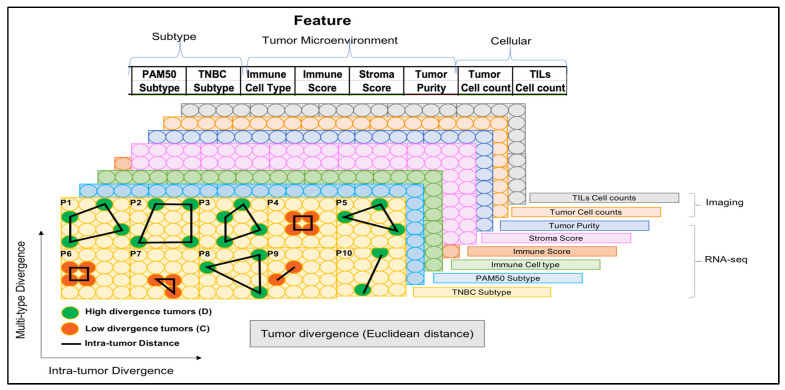
Summary of heterogeneity in features derived from gene expression and imaging data. According to the degree of variability in each feature, patients were stratified into patients with divergent tumors and those with convergent tumors.

**Figure 6 ijms-23-13322-f006:**
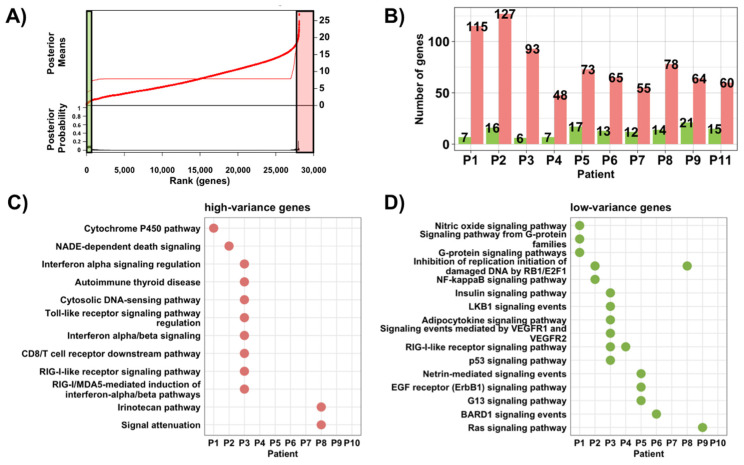
Identification of gene signatures with high and low regional gene expression heterogenetiy within patients. (**A**) Example of a Bayesian changepoint model to assess the distribution of gene distance measures to define genes with high (pink) and low expression variability (green). (**B**) Frequency distribution of the number of genes with high (pink) and low (green) ITH in their expression levels based on a Bayesian changepoint model. (**C**,**D**) Dot plot showing BioPlanet pathways significantly (*p* < 0.05) enriched for gene sets with high and low ITH. The dots are color-coded according to the *p*-value.

**Figure 7 ijms-23-13322-f007:**
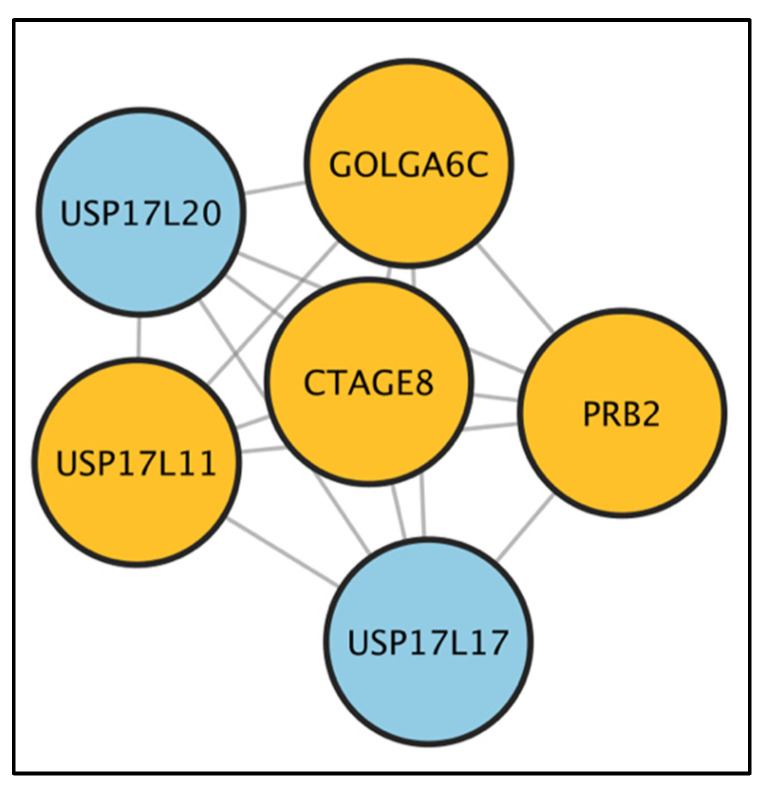
Enriched networks of highly variable, patient-specific genes in long-term survivors versus deceased patients. Protein-coding network of six significant (*p* = 0.11) submodules identified as enriched in long-term survivors versus deceased TNBC patients; seed genes are shown in orange.

**Table 1 ijms-23-13322-t001:** Clinicopathological characteristics of patients.

Menopausal Status, *n* (%)	
Pre	6 (60%)
Post	4 (40%)
Age at diagnosis, *n* (%)	
<50	4 (40%)
≥50	6 (60%)
Tumor size, *n* (%)	
<2.0	2 (20%)
≥2.0	8 (80%)
Grade, *n* (%)	
1	1 (10%)
2	1 (10%)
3	8 (80%)
LN stage, *n* (%)	
1	2 (20%)
2	5 (50%)
3	3 (30%)
Chemotherapy, *n* (%)	
CMF	7 (70%)
No therapy	1 (10%)
Missing Data	2 (20%)
Recurrence, *n* (%)	
Yes	4 (40%)
No	6 (60%)
Distant metastasis, *n* (%)	
Yes	4 (40%)
No	6 (60%)
Alive or dead, *n* (%)	
Alive	6 (60%)
Died from Breast Cancer	4 (40%)

**Table 2 ijms-23-13322-t002:** Summary of the impact of regional heterogeneity in gene expression-derived and image-derived features. “C” denotes convergence (i.e., low intratumor heterogeneity among regions) and “D” denotes divergence (i.e., high intratumor heterogeneity among regions). The “Divergent Association” row highlights patient and tumor associations with the divergent versus convergent results at *p* < 0.20 based on Fisher’s exact test.

Patient	RNA-Seq-Derived Gene Expression						Image-Derived Cell Counts	
	PAM50	TNBC	Immune Cell	Immune	Stroma	Tumor	Tumor	TILs
	Subtype	Subtype	Type	Score	Score	Purity	Cell Counts	Cell Counts
P1	D	D	D	D	D	C	D	D
P2	D	D	C	D	D	D		
P3	D	D	D	D	C	D	C	D
P4	D	C	D	D	C	D	C	D
P5	C	D	C	D	C	D	D	C
P6	D	C	C	C	C	C	D	D
P7	D	C	C	C	C	D	D	D
P8	D	D	D	C	C	C		
P9	D	C	D	C	C	C	C	C
P10	C	D	C	C	C	C		
Divergent	CMF	Lymph Node Stage	CMF		Lymph Node Stage			Grade
Association	Chemotherapy		Chemotherapy					

## Data Availability

The data presented in this study are available in this article (and Appendix A).

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
