# Peer review of "A Case Series Exploration of Multi-Regional Expression Heterogeneity in Triple-Negative Breast Cancer Patients"

_ijms, 2022, doi:10.3390/ijms232113322_

Round 1

Reviewer 1 Report

This is a very interesting work done at a high level.

I have no comments that would prevent the publication of this article.

I have a few wishes for the design of the article

1) Figure 3A - the text is hard to read

2) The text in figure 6 should be enlarged for better readability

Author Response

Please see the attachment-Cover_Letter_ITH_IJMS_Oct12

Reviewer 2 Report

This paper discusses the molecular heterogeneity within the same tumor and between patients diagnosed with triple-negative breast cancer. They emphasize that the identification of subtypes of triple-negative cancer is of importance, as it may help in the development of personalized treatments for each TNBC subtype.

Introduction:

It could be improved by adding information already reported related to TNBC heterogeneity, adding information related to fibroblast, stromal cells, and immune cells (resident and infiltrating)

Material and Methods:

At the end of the second line of page 6, change the word “pat4ients” to ”patients”.

On lines 7 and 8, authors must add a space between the words “TNBCtype”.

Results:

The authors mention that their work was carried out with 10 patients (P1-P10); however, in table 2 of the supplementary material, there is no P10, and they refer to a P11. It should be corrected or clarified, who is the P11.

The authors should define each subtype (BL, UNS, MSL, IM, M, LAR) in Figure 4C.

It could be helpful if results are rearranged according to variability between patients (Figures 2, 3, 6, and 7), and intra-tumoral variability (Figures 4, 5, and table 2)

Discussion

Although the authors did not identify significant enrichment of distinctive cancer gene sets in their gene signature among patients from all regions, the authors are encouraged to discuss the hallmark of immune evasion, as they found immunosurveillance-associated cells in their immunogenic signature, which could be related to the aggressiveness and recurrence of TNBC.

Although it is not the objective of this work to identify the different cell populations that coexist in the same tumor, it is convenient for the authors to discuss the relationship that the composition of the diverse cell populations may have since these may be the cause of intra-tumoral heterogeneity, as well as the presence of cancer stem cells (CSCs) since in their analysis they found a significant enrichment in the Notch and PD-1 pathways; also, this finding should be highlighted in the results section.

The authors should discuss the results related to pre-and post-menopausal status (Figure 2) and the pathways enrichment results (figure 3).

Author Response

(The authors gave the same response as above.)
